# ATF4 licenses C/EBPβ activity in human mesenchymal stem cells primed for adipogenesis

Daniel M Cohen[1,2], Kyoung-Jae Won[2,3], Nha Nguyen[2,3], Mitchell A Lazar[1,2,3], Christopher S Chen[4,5]*, David J Steger[1,2]*

[1]Division of Endocrinology, Diabetes, and Metabolism, Department of Medicine, Perelman School of Medicine, University of Pennsylvania, Philadelphia, United States; [2]The Institute for Diabetes, Obesity, and Metabolism, Perelman School of Medicine, University of Pennsylvania, Philadelphia, United States; [3]Department of Genetics, Perelman School of Medicine, University of Pennsylvania, Philadelphia, United States; [4]Department of Biomedical Engineering, Boston University, Boston, United States; [5]Wyss Institute for Biologically Inspired Engineering, Harvard University, Cambridge, United States

**Abstract** A well-established cascade of transcription factor (TF) activity orchestrates adipogenesis in response to chemical cues, yet how cell-intrinsic determinants of differentiation such as cell shape and/or seeding density inform this transcriptional program remain enigmatic. Here, we uncover a novel mechanism licensing transcription in human mesenchymal stem cells (hMSCs) adipogenically primed by confluence. Prior to adipogenesis, confluency promotes heterodimer recruitment of the bZip TFs C/EBPβ and ATF4 to a non-canonical C/EBP DNA sequence. ATF4 depletion decreases both cell-density-dependent transcription and adipocyte differentiation. Global profiling in hMSCs and a novel cell-free assay reveals that ATF4 requires C/EBPβ for genomic binding at a motif distinct from that bound by the C/EBPβ homodimer. Our observations demonstrate that C/EBPβ bridges the transcriptional programs in naïve, confluent cells and early differentiating pre-adipocytes. Moreover, they suggest that homo- and heterodimer formation poise C/EBPβ to execute diverse and stage-specific transcriptional programs by exploiting an expanded motif repertoire.

*For correspondence: chencs@bu.edu (CSC); stegerdj@mail.med.upenn.edu (DJS)

**Competing interests:** The authors declare that no competing interests exist.

## Introduction

Post-mitotic adipocytes arise from the differentiation of MSCs in a process termed adipogenesis (*Cristancho and Lazar, 2011*; *Rosen and Spiegelman, 2014*). The cell-fate decisions occurring during the stages of adipogenesis are controlled by multiple sequence-specific transcription factors (TFs) (*Farmer, 2006*; *Rosen and MacDougald, 2006*; *Cristancho and Lazar, 2011*). Best described among these are PPARγ and C/EBP TFs, which drive a process of terminal differentiation that results in the expression of metabolic genes and adipokines important for the adipocyte phenotype (*Hwang et al., 1997*; *Rosen and MacDougald, 2006*; *Lefterova and Lazar, 2009*). PPARγ is recognized as a master regulator because it is necessary (*Barak et al., 1999*; *Kubota et al., 1999*; *Rosen et al., 1999*) and sufficient (*Tontonoz et al., 1994*; *Hu et al., 1995*; *Shao and Lazar, 1997*) for adipogenesis. C/EBP proteins can also induce adipocyte differentiation of fibroblasts, although none can induce differentiation in the absence of PPARγ (*Rosen et al., 2002*). Cistromic analyses have revealed that PPARγ and C/EBPα occupy sites near most of the genes that are up-regulated during adipogenesis, including their own, suggesting that they coordinate expression of the majority of genes determining adipocyte function

**eLife digest** Human body fat consists mostly of fat-storing cells called adipocytes. These cells develop in two main steps. First, mesenchymal stem cells—which can potentially become one of many different types of cell—commit to becoming pre-adipocyte cells. These pre-adipocytes then develop into mature adipocytes. Proteins called transcription factors control both steps of this process by binding to particular sites in the DNA of the cell and activating certain genes that control the cell's identity and activity.

Various different transcription factors are known to stimulate the development of mesenchymal stem cells into adipocytes. Experiments performed on cells that have been grown in the laboratory suggest that the cells must also be tightly packed together to become adipocytes.

Cohen et al. have now investigated the role a protein called C/EBPβ plays in the development of adipocytes, and have found that it plays different roles at different stages of development. When mesenchymal stem cells become tightly packed, more of another protein called ATF4 is produced. This protein binds to C/EBPβ, and the resulting two-protein complex then binds to sites on the DNA of the mesenchymal stem cell to activate genes that turn the stem cell into a pre-adipocyte. Reducing the amount of ATF4 in mesenchymal stem cells reduces the number of pre-adipocytes that develop.

When not bound to ATF4, and in response to certain cell signals, C/EBPβ binds to different DNA sites and helps pre-adipocytes develop into mature adipocytes. Whether transcription factors other than ATF4 partner with C/EBPβ to change DNA-binding-site preferences remains unknown. Future studies will search for these, with the aim of providing a clearer molecular understanding for how C/EBPβ acts as a 'bridge' that links together the two stages of adipocyte development.

(*Lefterova et al., 2008*; *Nielsen et al., 2008*; *Mikkelsen et al., 2010*; *Schmidt et al., 2011*; *Soccio et al., 2011*). Thus, the transcriptional circuitry formed by PPARγ and C/EBPα may explain how adipocyte cell identity is established and maintained.

Genomics studies have provided important new insights into the TFs acting before PPARγ and C/EBPα. Treatment of pre-adipocyte fibroblasts with a cocktail including glucocorticoid, cyclic AMP agonists and insulin (DMI) and activates thousands of putative enhancers with increased sensitivity to DNase I, enrichment for activating histone modifications, and induced binding by TFs (*Mikkelsen et al., 2010*; *Siersbæk et al., 2014a*). Acting transiently during the early stages of adipogenesis, these TFs appear to serve at least two important functions: they recruit co-activators such as p300 and MED1 to activate genes important for terminal differentiation including *PPARγ* (*Steger et al., 2010*; *Siersbæk et al., 2014b*), and they remodel chromatin to facilitate later binding by PPARγ and C/EBPα in adipocytes (*Siersbæk et al., 2011*).

While DMI is a potent trigger for adipogenesis in both mouse 3T3-L1 fibroblasts and human mesenchymal stem cells (hMSCs), it is effective only when cells are densely packed (*Green and Kehinde, 1976*; *Pittenger et al., 1999*; *McBeath et al., 2004*; *Cristancho et al., 2011*). Here, we use this cell density 'checkpoint' as a means to identify transcriptional mechanisms that prime cells to a permissive pre-adipogenic state. We identify on a genome-wide scale putative enhancer and promoter regions with differential occupancy for RNA Polymerase II (RNAPII) in response to changes in cell density and treatment with DMI cocktail. The findings support roles for C/EBPβ and GR as primary drivers of DMI-induced gene expression.

We also observe a surprising enrichment for C/EBPβ-binding sites at RNAPII enhancers responding to high-seeding density prior to the addition of DMI cocktail. While lacking a canonical, palindromic C/EBPβ motif, these enhancers exhibit dramatic enrichment for an asymmetric, composite motif that juxtaposes half-sites for canonical C/EBP and AP-1 motifs. We demonstrate that these hybrid motifs recruit C/EBPβ as a heterodimer with another bZIP family member, ATF4. Genome-wide binding by ATF4 demonstrates its exclusive co-localization with C/EBPβ, and depletion of ATF4 decreases adipogenesis. Together, these observations suggest that a program of C/EBPβ-ATF4-dependent gene expression triggered by high-seeding density plays an important role in priming hMSCs for adipogenesis. The observation that C/EBPβ hetero- and homodimeric complexes exhibit different sequence specificities provides novel mechanistic insights into how C/EBPβ can be differentially targeted to control distinct programs of gene expression at distinct phases of adipocyte differentiation,

and revises the prevailing view that C/EBPβ is transcriptionally inactive in the absence of exogenous adipogenic stimuli (*Wiper-Bergeron et al., 2003*; *Raghav et al., 2012*).

## Results

### Identification of regulated enhancers during hMSC differentiation

RNAPII is recruited to active enhancers on a global scale (*Szutorisz et al., 2005*; *Koch et al., 2008*; *Kim et al., 2010*), and we performed RNAPII ChIP-seq in primary hMSCs to annotate putative cis-acting regulatory elements during human adipocyte differentiation. Using a feature detection algorithm that robustly identifies enriched regions, we captured a progression of differentiated states when cells were cultured either at non-permissive, low density (LD) or permissive, high density (HD) in the presence or absence of DMI adipogenic cocktail (*Figure 1A*). K-means clustering uncovered differential RNAPII occupancy that fell broadly into three categories: associated with LD hMSCs (clusters 1–8, uncommitted), preferentially associated with HD hMSCs (clusters 9–12, primed), and associated with DMI induction (clusters 13–16). Clusters 17–19 displayed an ambiguous relationship to adipocyte differentiation and were excluded from subsequent analyses.

The uncommitted, HD-primed, and DMI-induced RNAPII clusters mapped to distinct gene ontologies (GOs) (*Figure 1B*, *Supplementary file 1*). Cytoskeleton and cell proliferation genes were enriched in the uncommitted clusters, and genes associated with biological processes characteristic of terminally differentiated adipocytes, namely PPAR signaling, pyruvate and lipid metabolism, and adipocytokine signaling were only enriched when the cells were cultured at high-seeding density and treated with DMI cocktail. Of particular note, genes involved in amino acid transport and metabolism were associated with the HD-primed clusters, where high-seeding density promoted RNAPII recruitment prior to DMI induction.

Intriguingly, the HD-primed, but not the uncommitted and DMI-induced, RNAPII sites associated with a CHOP/ATF4/hybrid motif (*Figure 1C*, *Figure 1—figure supplement 1*). It is a hybrid of AP-1 (TGA) and C/EBP (TTGC) half-sites (*Figure 1D*), and binds heterodimers between different C/EBP proteins and AP-1 or ATF factors (*Vinson et al., 1993*; *Shuman et al., 1997*; *Wolfgang et al., 1997*; *Cai et al., 2008*). ChIP-seq for C/EBPβ, which is known to control early phases of adipogenesis in murine cells along with GR (*Steger et al., 2010*; *Siersbæk et al., 2011*), revealed extensive binding in HD hMSCs that selectively mapped to the HD-primed RNAPII sites (*Figure 1A*, track 5, *Figure 1—figure supplement 2A,B*). Treatment of HD cultures with DMI cocktail led to a massive up-regulation of C/EBPβ genomic occupancy (*Figure 1—figure supplement 2A*). It also induced binding by GR that mapped to regions with DMI-induced RNAPII (*Figure 1A* and *Figure 1—figure supplement 2B*) and co-localized with C/EBPβ (*Figure 1—figure supplement 2C*) in a density-independent manner (*Figure 1—figure supplement 1D,E*). These data indicate that C/EBPβ targets HD-primed enhancers before, and DMI-induced enhancers during, human adipogenesis.

### C/EBPβ regulates HD-primed enhancers through a non-canonical motif

To test whether C/EBPβ drives the activity of HD-primed enhancers via a non-canonical binding site as predicted by the motif analysis of RNAPII sites, we screened C/EBPβ-binding regions at all RNAPII-annotated regions for the presence of a canonical CEBP sequence (TTGCnnAA) (*Jolma et al., 2013*) or a hybrid motif (TTKCATCA) (*Figure 2A*). Whereas the hybrid motif is present at 27% of C/EBPβ peaks in the HD-primed clusters, only 7% and 4% of peaks have this sequence in the uncommitted and DMI-induced clusters, respectively, indicating its enrichment within the C/EBPβ-binding sites of HD-primed enhancers. Conversely, 23% of C/EBPβ peaks in the DMI-induced clusters are associated with the canonical C/EBP motif, compared to 13% and 8% of peaks in the uncommitted and HD-primed clusters, respectively. These results demonstrate that C/EBPβ preferentially targets the hybrid motif in the HD-primed enhancers, whereas it binds the canonical C/EBP motif in the DMI-induced enhancers.

The HD-primed and DMI-induced enhancers are active during distinct stages of differentiation, and employ different motifs to recruit C/EBPβ, suggesting that distinct classes of C/EBPβ-recognition sequences confer temporal regulation of gene transcription during adipocyte development. To address this idea, we placed several RNAPII-annotated regions that are targeted by C/EBPβ and harbor either a hybrid or canonical motif into luciferase reporters, and assayed their activity in response to DMI cocktail (*Figure 2B*). Interestingly, reporters carrying the hybrid motif had reduced in activity in the presence of DMI, while those with the canonical C/EBP sequence were increased by DMI.

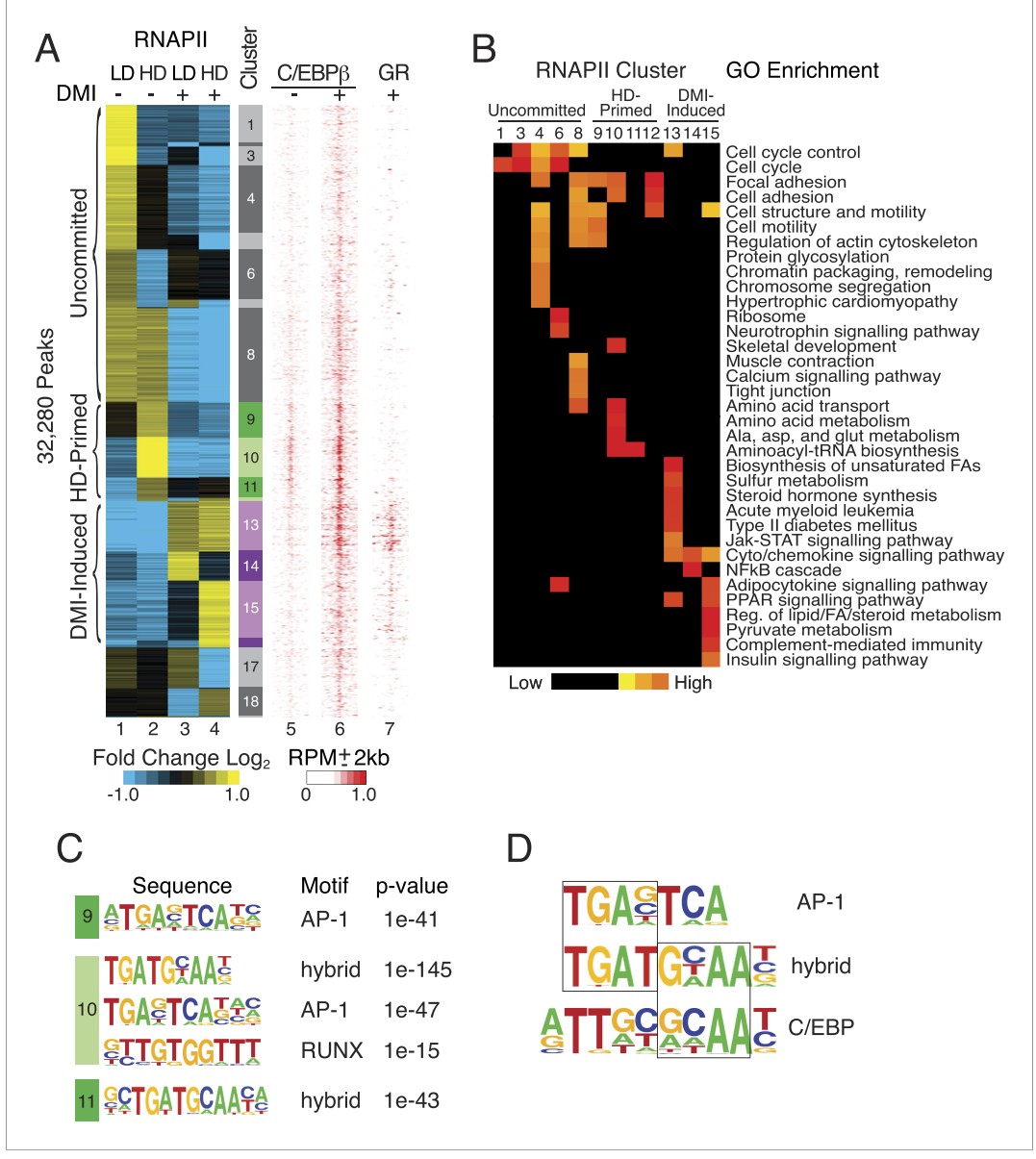

**Figure 1**. RNAPII-annotated enhancers reveal stage-specific transcriptional programs during adipogenic commitment. (**A**) Heat map of RNA Polymerase II (RNAPII) peaks clustered as a function of differential enrichment in response to changes in cell density (low, LD vs high, HD) and differentiation cocktail (+ or − DMI, 24 hr) in human mesenchymal stem cells (hMSCs) (tracks 1–4). Density heat maps (red) of C/EBPβ (0 and 6 hr DMI) and GR (6 hr DMI) binding in hMSCs within a 4 kb window of differential RNAPII peaks (tracks 5–7). Color bar specifically refers to scaling for C/EBPβ tracks. (**B**) Gene ontologies (GOs) resulting from mapping RNAPII enhancers to genes with correlated changes in gene body RNAPII. (**C**) Identification of de novo motifs in differential RNAPII clusters 9–11; reported sequences met detection thresholds of at least 5% of targets and p-value ≤ 1e-14. (**D**) Comparison of the AP-1, hybrid and C/EBP motifs, with the AP-1- and C/EBP-half sites boxed.

The following figure supplements are available for figure 1:

**Figure supplement 1**. Characterization of RNAPII-annotated enhancers.

**Figure supplement 2**. C/EBPβ and GR ChIP-seq in hMSCs.

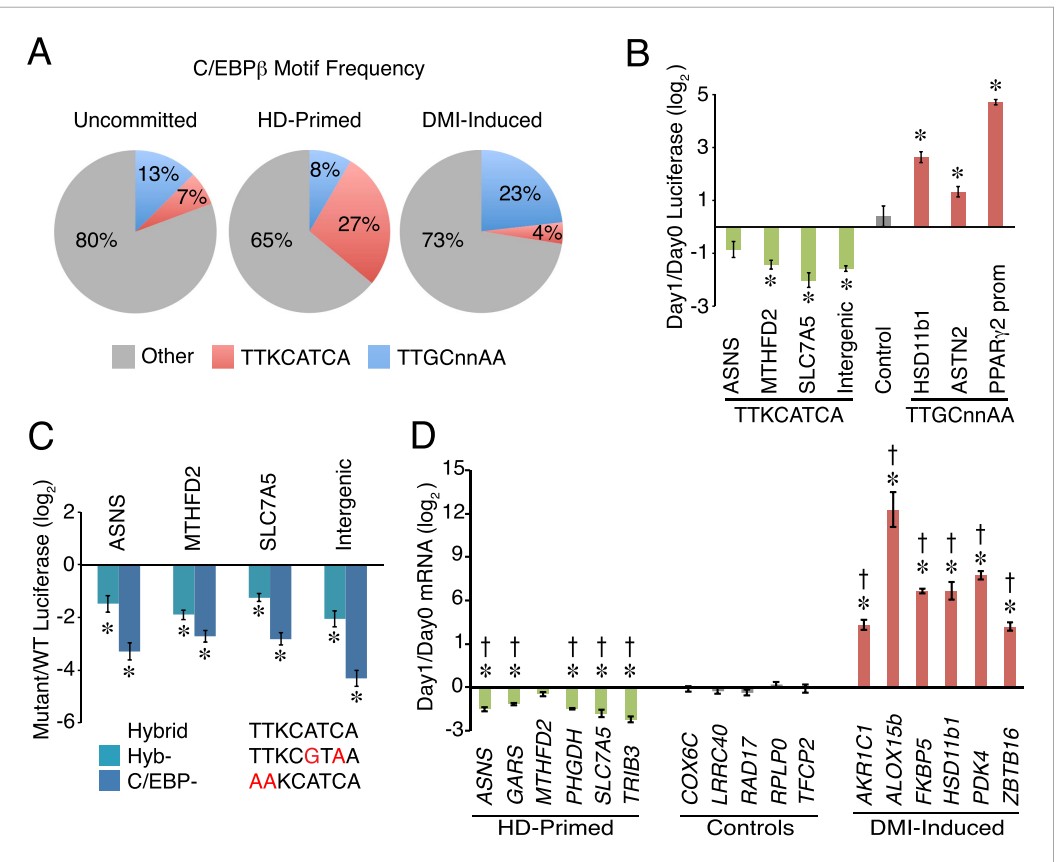

**Figure 2**. A non-canonical C/EBP motif mediates transcription at HD-primed enhancers. (**A**) Pie charts comparing the frequency of hybrid (red) vs canonical C/EBP (blue) motifs present at RNAPII-C/EBPβ co-bound sites at stage-specific enhancers. (**B**) C/EBPβ enhancers harboring hybrid (green) or canonical C/EBP (red) motifs were assayed by a luciferase reporter in the absence (day 0) or presence (day 1) of DMI cocktail in C3H10T1/2 cells. (**C**) Mutations disrupting the hybrid motif specifically (cyan) or C/EBP motif generally (blue) were assayed by a luciferase reporter in basal medium in C3H10T1/2 cells. (**D**) Gene expression was assayed for genes associated with C/EBPβ enhancers containing hybrid (green) or canonical C/EBP motifs (red), and in a panel of endogenous control genes (gray), in DMI-treated (day 1) vs untreated (day 0) hMSCs. *, denotes p < 0.05, Student's *t*-test comparing stimulated vs unstimulated or mutant vs WT; †, denotes p < 0.002, comparison of fold changes between stage-specific enhancers vs endogenous controls, Mann–Whitney test. Error bars depict SEM.

Moreover, the hybrid motif mediates transcriptional activity since targeted mutation of either the hybrid sequence specifically (Hyb-) or the C/EBP motif generally (C/EBP-) lowered luciferase activity (*Figure 2C*). To further test the functional significance of C/EBP motifs in regulating gene transcription, we quantified mRNAs for genes with nearby RNAPII-annotated regions bound by C/EBPβ (*Figure 2D*). Consistent with the changes in both RNAPII occupancy and the luciferase reporters, genes associated with the hybrid motif showed decreased expression upon DMI stimulation. In contrast, genes associated with the canonical C/EBP motif (*AKR1C1*, *ALOX15b*, *FKBP5*, *HSD11b1*, and *ZBTB16*) were activated by DMI treatment along with a control adipocyte marker (*PDK4*). As a whole, these findings suggest that C/EBPβ targets a non-canonical C/EBP motif to activate HD-primed enhancers. Furthermore, although the addition of DMI cocktail increases C/EBPβ occupancy globally, including sites associated with the HD-primed enhancers, it reduces the activity of the HD-primed enhancers while inducing enhancers harboring canonical C/EBP motifs.

## ATF4 is a density-dependent factor that targets HD-primed enhancers as a heterodimer with C/EBPβ

Hybrid motifs recruit C/EBP TFs as heterodimers with bZip proteins from the AP-1 or ATF families (*Vinson et al., 1993*; *Shuman et al., 1997*; *Wolfgang et al., 1997*; *Cai et al., 2008*).

Therefore, we performed ChIP to measure the occupancy of several candidate partners for C/EBPβ, and found robust binding of ATF4 at hybrid motifs (*Figure 3A* and *Figure 3—figure supplement 1A*). To map ATF4-binding sites on a genome-wide scale, we performed ChIP-seq in pre-adipocyte hMSCs (HD, without DMI), and identified 1451 discrete ATF4 peaks. Remarkably, ATF4 binds almost exclusively with C/EBPβ, with more than 90% of its sites co-localized with C/EBPβ-binding sites (*Figure 3B*). Moreover, motif analysis shows an extraordinary level of sequence specificity at ATF4 binding sites, with strong nucleotide selection at nearly every position of the core hybrid motif, and extremely high concordance between the presence of ATF4 and the hybrid motif, with 85% of peaks containing the recognition sequence (*Figure 3C*). In marked contrast, the C/EBPβ cistrome yields

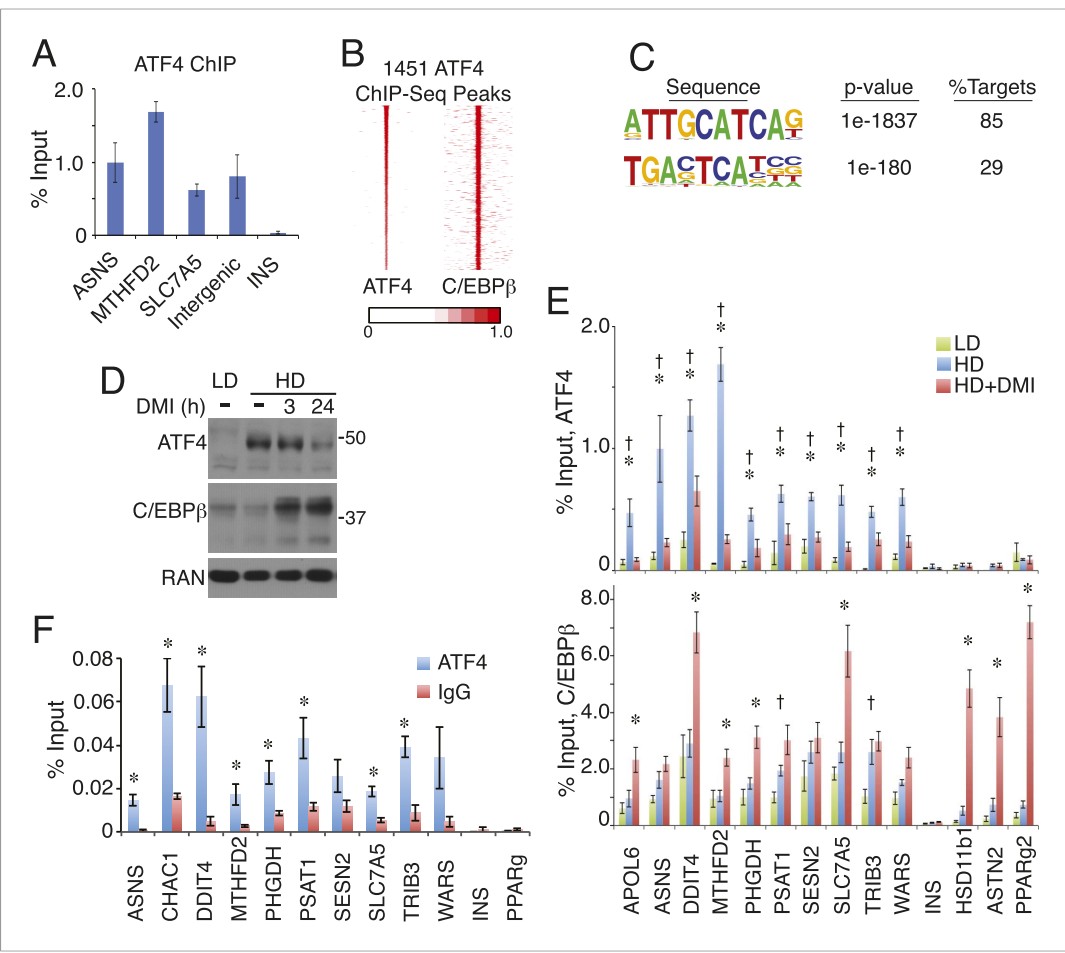

**Figure 3**. ATF4 is a density-regulated factor that binds to HD-primed enhancers as a heterodimer with C/EBPβ. (**A**) ATF4 ChIP in high density (HD) hMSCs interrogating genomic loci that contain the hybrid motif vs non-specific control (INS). (**B**) Density heat map anchored on ATF4 ChIP-seq peaks to examine co-distribution with C/EBPβ on a genome-wide scale in HD hMSCs. (**C**) De novo motif analysis of ATF4 ChIP-seq peaks. (**D**) Changes in ATF4 and C/EBPβ protein levels assessed by Western blotting during adipogenic differentiation. MW markers, 50 and 37 kDa. (**E**) Bar graphs of ATF4 (top) and C/EBPβ (bottom) occupancy at hybrid or canonical motifs in hMSCs cultured at low (green bars) or high (blue bars) seeding density in the absence of DMI or following 24 hr of treatment (red bars). *, denotes $p < 0.05$, Student's $t$-test comparison of HD samples with vs without DMI treatment; †, denotes $p < 0.05$, Student's $t$-test comparison of LD vs HD samples without DMI treatment. (**F**) Sequential ChIP of C/EBPβ followed by ATF4 or IgG to interrogate simultaneous occupancy of both factors at hybrid motifs. *, denotes $p < 0.05$ for a one-tailed Student's $t$-test. Error bars depict SEM.

The following figure supplement is available for figure 3:

**Figure supplement 1**. Characterization of ATF4-binding sites.

a motif with a higher degree of sequence degeneracy and a lower detection frequency (65%) at binding sites (*Figure 1—figure supplement 2A*).

Genomic occupancy of ATF4 is associated predominantly with genes involved in tRNA amino-acylation for protein synthesis, response to endoplasmic reticulum (ER) stress, and the unfolded protein response (*Figure 3—figure supplement 1B*). ATF4 occupies sites near all of the genes observed in the HD-primed clusters that function in amino acid transport and metabolism and tRNA aminoacylation (*Supplementary file 1*). Relative to C/EBPβ, ATF4 occupies sites that are highly enriched for RNAPII, with the majority of these falling in the HD-primed clusters (*Figure 3—figure supplement 1C,D*). Thus, ATF4 is intimately tied to HD-primed RNAPII, implicating it in the activation of gene transcription in response to high-cell-density seeding of hMSCs. These data also imply that ATF4 abundance may be regulated by cell density. Indeed, western analysis revealed that while ATF4 levels in uncommitted LD cells were negligible, ATF4 was up-regulated in response to high-cell-seeding density (*Figure 3D*). Furthermore, upon addition of DMI cocktail, ATF4 expression diminished modestly at 3 hr and prominently by 24 hr. As expected, C/EBPβ showed a rapid and sustained induction by DMI in hMSCs. To determine if ATF4 genomic occupancy was correlated with its abundance, we performed ChIP in cells cultured at LD or HD with or without DMI treatment. ATF4 occupancy at hybrid sites was strongly reduced in either untreated LD hMSCs or DMI-treated HD hMSCs (*Figure 3E*). C/EBPβ, in contrast, bound to the hybrid sites irrespective of the cell seeding conditions. Concomitant with C/EBPβ up-regulation by differentiation cocktail, a trend of increased binding was apparent in the DMI condition. These observations indicate that ATF4 protein level is modulated by signals derived from both cell density and DMI cocktail, leading to functional interaction with C/EBPβ in primed hMSCs as well as early in adipogenesis.

Essentially no ATF4 peaks were observed that lack co-bound C/EBPβ, suggesting heterodimer binding. To interrogate whether ATF4 and C/EBPβ can simultaneously occupy the same genomic loci, we performed sequential ChIP experiments (*Figure 3F*). After an initial ChIP for C/EBPβ, ATF4 binding was highly enriched relative to the IgG control at all sites containing a hybrid motif, while no such enrichment occurred at control sites where C/EBPβ binds independently of ATF4 (PPARγ) or is absent (INS). Together, the data demonstrate ATF4 is a density-dependent TF that co-localizes with HD-primed enhancers in hMSCs by targeting the hybrid motif as a heterodimer with C/EBPβ.

## Loss of ATF4 attenuates adipogenesis and cell-density-dependent transcription

To investigate whether ATF4 regulates adipocyte differentiation, we depleted its expression in primary hMSCs, and in the well-characterized mouse pre-adipocyte cell line 3T3-L1 (*Figure 4A*). Transient transfection with ATF4 siRNAs decreased the level of Oil Red O staining relative to controls, indicating lower lipid accumulation in the ATF4 knockdown cells (*Figure 4B*). In parallel, early markers of adipogenesis were decreased threefold–fourfold in ATF4 knockdown cells (*Figure 4C*), including the mRNAs encoding the lineage-determining TFs PPARγ2 and C/EBPα, as well as several of their downstream transcriptional targets. This deficit in adipogenic gene expression became more pronounced over time (*Figure 4—figure supplement 1*), and together, these data suggest that ATF4 expression is important for adipogenesis. To better define the role of ATF4, we measured genomic binding in ATF4 knockdown cells. As expected, ATF4 occupancy was significantly reduced at all sites (*Figure 4D*). Notably, RNAPII occupancy was diminished at all but one of the ATF4-binding sites, and most of these showed a significant reduction, indicating that ATF4 functions to recruit RNAPII. C/EBPβ binding was decreased at all sites, but reached significance at a minority, which is not surprising given that C/EBPβ can occupy these sites under conditions with little or no ATF4 binding. These data establish ATF4 as a novel density-dependent TF that transduces signals from multiple pro-adipogenic stimuli including cell density and DMI cocktail to program the activity of HD-primed enhancers and promote adipocyte differentiation.

## ATF4, C/EBPβ and the C/EBPβ-ATF4 heterodimer have distinct DNA-binding specificities

Our data suggest a model for density-dependent gene expression whereby C/EBPβ-ATF4 heterodimers activate hybrid-motif-bearing enhancers in response to increased cell density by recruiting RNAPII. This stage-specific gene expression program is driven by robust sequence-specificity of

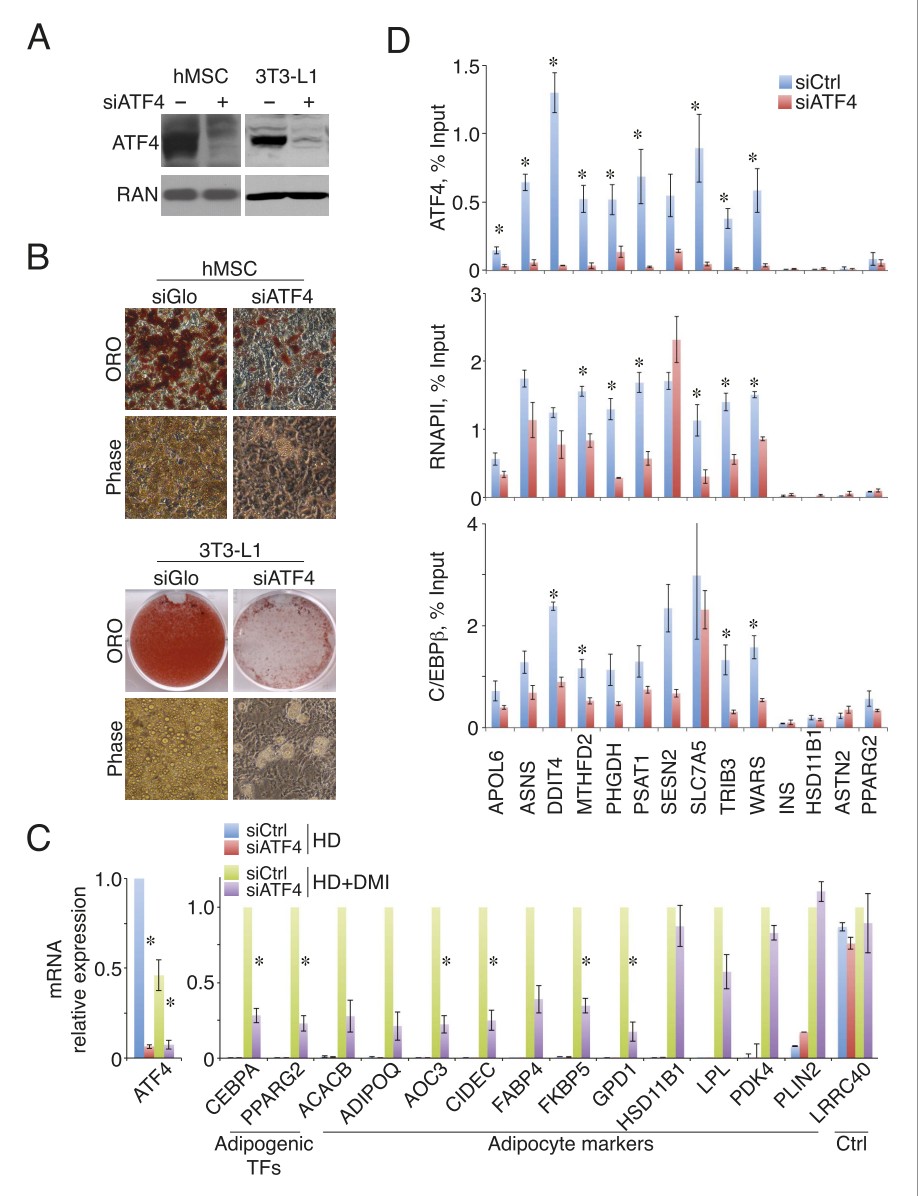

**Figure 4**. ATF4 promotes adipocyte differentiation and density-dependent transcription. (**A**) ATF4 knockdown in hMSCs and 3T3-L1 pre-adipocytes assessed by western blotting. RAN, loading control. (**B**) hMSCs (top) and 3T3-L1 cells (bottom) were transfected with siATF4 RNA duplexes or control sequences, and assessed for lipid droplet formation following 1 week of adipogenic differentiation. Top images, Oil Red O staining at the level of individual cells (hMSCs) or for an entire well of a 6-well plate (3T3-L1s). Bottom panel, phase contrast images. (**C**) HD hMSCs were transfected with control (PPIB) or ATF4 siRNAs and assayed for mRNA expression of early adipogenic markers at days 0 (HD) and 3 (HD+DMI) of differentiation. Expression was normalized to the maximum level for the control siRNA. (**D**) Changes in ATF4 (top), RNAPII (middle) and C/EBPβ (bottom) occupancy as a function of ATF4 knockdown assessed by ChIP in undifferentiated hMSCs cultured at high seeding density. *, denotes p < 0.05. Error bars depict SEM.

The following figure supplement is available for figure 4:

**Figure supplement 1**. Gene expression in ATF4 knockdown cells.

the C/EBPβ-ATF4 heterodimer, such that it preferentially occupies hybrid sequences relative to the C/EBPβ homodimer, with ATF4's genomic function tied exclusively to C/EBPβ. Whether these findings are emergent properties of the intrinsic DNA-binding specificities of ATF4, C/EBPβ and the C/EBPβ-ATF4 heterodimer or result from other mechanisms is unknown. To gain insight into this, we performed EMSA with DNA probes representing the different motifs (*Figure 5A*), and recombinant ATF4 and C/EBPβ purified from *Escherichia coli* to ensure homogenous preparations that lacked other potential bZip binding partners typically present in mammalian cell extracts (*Figure 5B*). ATF4 exhibited no detectable binding to either sequence by itself, while C/EBPβ bound both, but preferred the palindromic C/EBP motif (*Figure 5C,D*). Addition of recombinant ATF4 to C/EBPβ increased binding to the ATF4 motif at the lower protein concentrations (*Figure 5C*, lanes 4–9), indicating that ATF4 and C/EBPβ readily form heterodimers in solution as shown previously for other bZip proteins (*Cao et al., 1991*), and revealing that the heterodimer recognizes this sequence with higher affinity than the C/EBPβ homodimer. In contrast, ATF4 decreased C/EBPβ occupancy of the palindromic C/EBP motif (*Figure 5D*, lanes 1–6), suggesting that the C/EBPβ-ATF4 heterodimer has lower affinity relative to the C/EBPβ homodimer for this sequence. Similar results were obtained with a reverse titration scheme (*Figure 5—figure supplement 1*), and

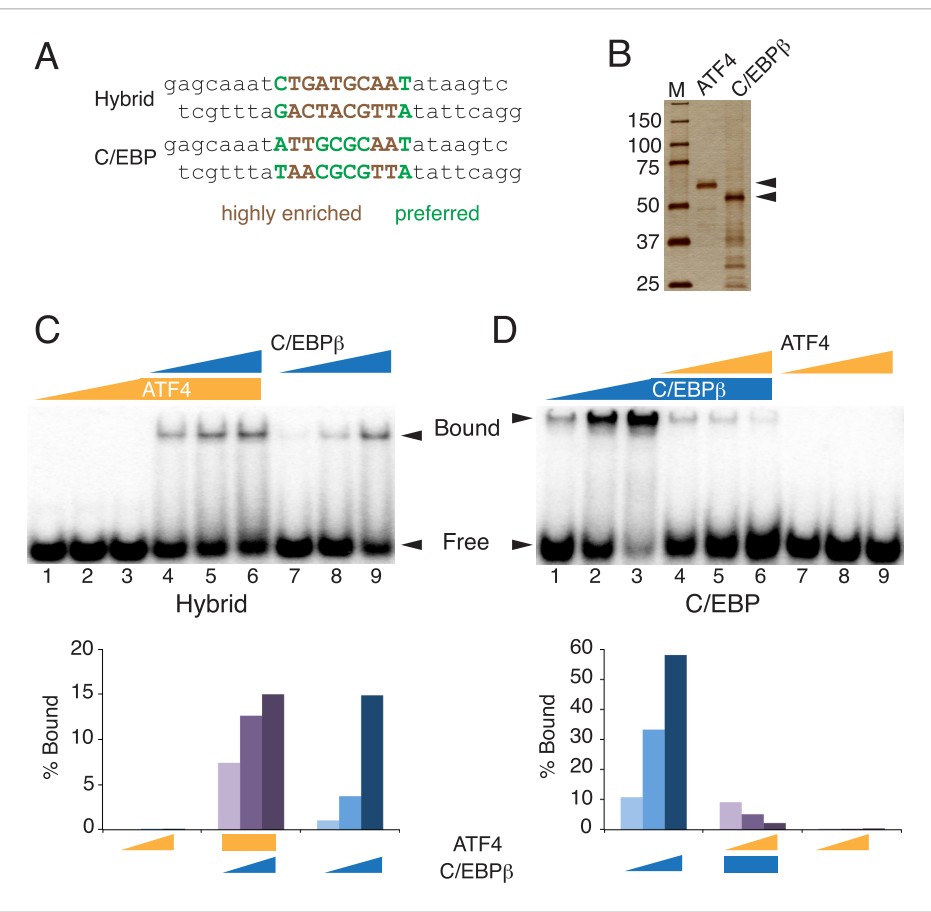

**Figure 5**. Distinct DNA-binding activities for ATF4, C/EBPβ and the C/EBPβ-ATF4 heterodimer. (**A**) EMSA probe design for the hybrid and C/EBP sequences. Motifs are capitalized. (**B**) Silver-stained SDS-PAGE of recombinant his-tagged ATF4 and C/EBPβ. Arrowheads indicate the expected MWs for the tagged constructs. (**C** and **D**) EMSA titration of ATF4 (10, 30 and 90 μM) and C/EBPβ (100, 300 and 900 nM) with 0.25 nM of either the hybrid (**C**) or C/EBP (**D**) radiolabeled probe. Bar graphs show phosphoimager-based quantification of bound complexes relative to free probe.

The following figure supplement is available for figure 5:

**Figure supplement 1**. DNA-binding activities for ATF4, C/EBPβ and the C/EBPβ-ATF4 heterodimer.

antibody controls demonstrated that ATF4 and/or C/EBPβ produce the protein-DNA complexes, further supporting these conclusions.

To extend our findings beyond two synthetic DNA templates, we tested the DNA-binding specificities of C/EBPβ and ATF4 to native sequences comprising the entire human genome. Genomic DNA was purified from hMSCs, stripped of protein, and sonicated to a relatively homogenous population of 250 bp fragments. Akin to EMSA, recombinant TFs and isolated DNA were interrogated via solution phase binding. Samples were processed subsequently in a manner similar to ChIP-seq using immunoprecipitation to isolate non-crosslinked protein-DNA complexes followed by analysis of bound DNA by next generation sequencing. A similar in vitro cistromics assay (IVC) has successfully examined TF binding with *Drosophila* components (*Guertin et al., 2012*). Here, we interrogated how TF concentration and dimerization state affect genomic occupancy in the absence of the confounding effects encountered in cells.

Titration of C/EBPβ over two orders of magnitude produced cistromes with robust peaks (*Figure 6A*) that scaled with protein amount (*Figure 6B*). The total number of bound sites exceeding a 1 RPM threshold plateaued near the low end of the C/EBPβ titration, and gained sites at the upper end of the titration had lower binding strength (*Figure 6C*), indicating that the addition of more protein drives binding to low-affinity DNA sites. In support of this, de novo motif analyses revealed stronger enrichment of the C/EBP motif at the top-ranked sites (*Figure 6D*). Yet, the C/EBP motif was the top-ranked sequence for both the strongest and weakest sites, indicating appropriate specificity for the C/EBPβ homodimer even at the higher protein levels. We anticipated that the number of C/EBPβ-binding sites observed in vitro would exceed that found in cells due to the removal of accessibility barriers imposed by chromatin. To test this, we examined the level of histone H3 lysine 27 acetylation (H3K27ac), a marker of active transcription and open chromatin (*Rada-Iglesias et al., 2011*; *Stasevich et al., 2014*), in hMSCs at regions bound by C/EBPβ (*Figure 6—figure supplement 1A*). Unlike sites occupied in hMSCs, the in-vitro-specific sites have little or no H3K27ac in cells, suggesting that they reside in inactive chromatin. Taken together, these results demonstrate that vitro cistromics interrogates protein-DNA specificity effectively.

To investigate the effects of heterodimerization on the global genomic distribution of C/EBPβ, we titrated ATF4 into the IVC. Upon addition of ATF4, C/EBPβ exhibited reduced occupancy of C/EBPβ-homodimer sites and de novo binding at sites not represented in the homodimer cistromes generated with higher C/EBPβ concentrations (*Figure 6E* and *Figure 6F*, tracks 1–4). Induced C/EBPβ-binding sites displayed parallel enrichment for ATF4 occupancy (tracks 5 and 6). Motif analysis showed robust enrichment of the hybrid motif at these sites, reproducing the exquisite specificity mapped in the hMSC ATF4 cistrome. Remarkably, ATF4 by itself exhibited no detectable binding to these regions (track 7), demonstrating that its interaction with the genome is exclusively heterodimeric. In total, we identified 111 peaks in the ATF4 homodimer cistrome, and these were enriched for multimers of a c-JUN motif (*Figure 6—figure supplement 1B,C*). These rare sites are not occupied by ATF4 in hMSCs, revealing that homodimeric binding by ATF4 may not be physiological. Together, the EMSA and in vitro cistromics experiments demonstrate that the DNA-binding properties of ATF4 and C/EBPβ are sufficient to reconstitute their sequence specificities observed in cells.

## Discussion

Our data reveal that C/EBPβ controls distinct gene expression programs in hMSCs at different stages of adipogenesis (*Figure 7*). Prior to treatment with DMI cocktail, high seeding density induces ATF4, which heterodimerizes with C/EBPβ, driving a unique set of genes that may prime hMSCs into a pre-adipocyte state. In contrast, in response to exogenous adipogenic signals, C/EBPβ activates a different set of genes, in part by co-localizing with GR. GR promotes adipocyte differentiation in mouse cells (*Steger et al., 2010*; *Asada et al., 2011*), and its association with induced genes during early hMSC adipogenesis suggests that it functions similarly in human cells.

This study identifies ATF4 as a cell density sensor that links the physical and chemical requirements for adipogenesis. High cell density has long been known to be an important condition for adipocyte differentiation in vitro (*Green and Kehinde, 1975*; *Pittenger et al., 1999*; *McBeath et al., 2004*; *Cristancho et al., 2011*). It is likely that a combination of cues is required in tissue microenvironments supporting adipogenesis in vivo. ATF4 joins a small, yet growing list of TFs that confer adipogenic competency. A major block in identifying TFs controlling the formation of pre-adipocyte cells is the lack of molecular markers distinguishing naïve hMSCs from committed pre-adipocytes.

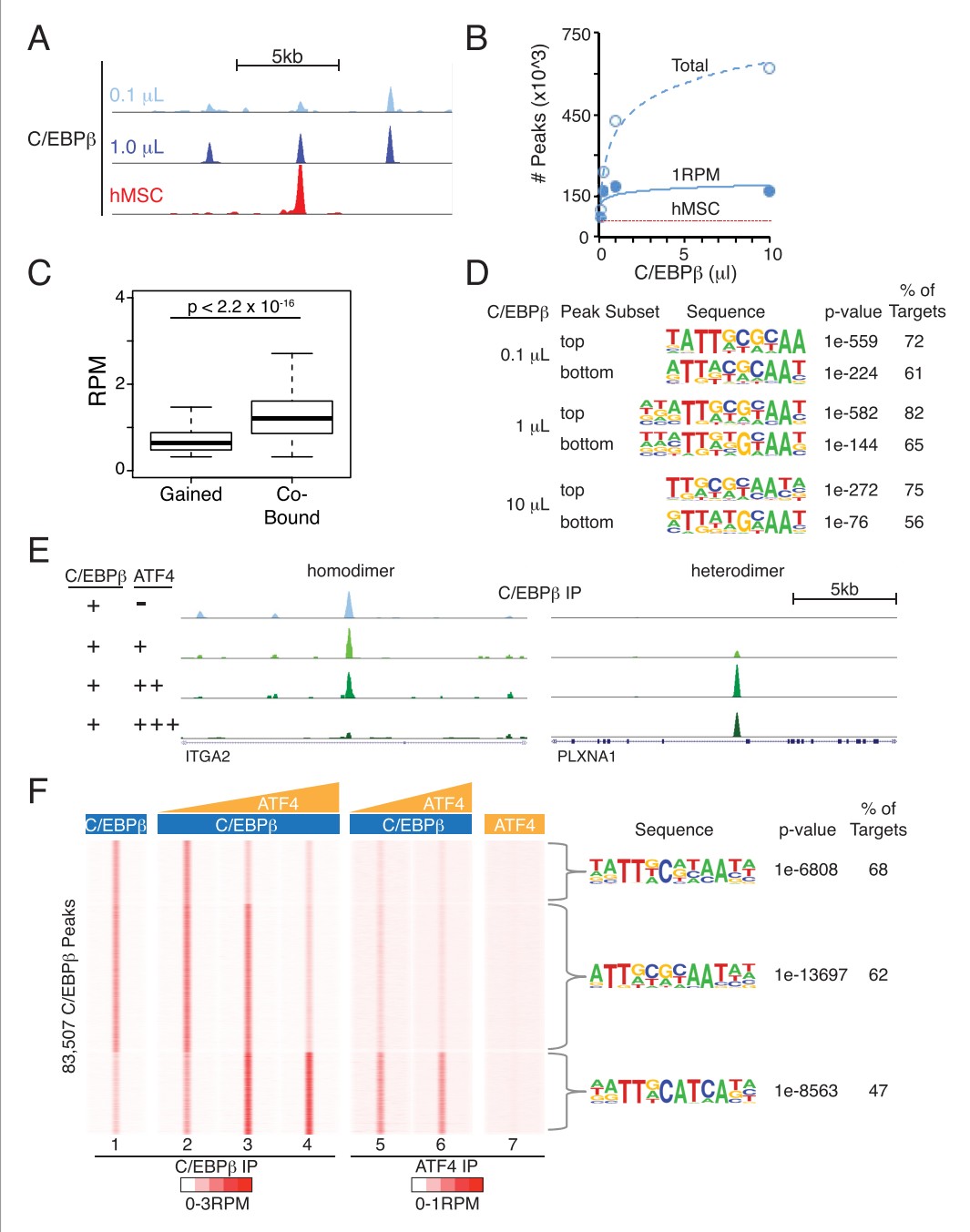

**Figure 6**. Heterodimer formation with C/EBPβ is necessary for ATF4 binding and is sufficient to alter C/EBPβ-sequence specificity. (**A**) Browser shot comparing C/EBPβ peaks from the in vitro cistromics assay (IVC) and hMSC ChIP-seq. 0.1 and 1 μl C/EBPβ molarity equivalents in the initial binding reaction are ~60 and 600 nM, respectively. Tracks are RPM normalized, y-axes scaled from 0–3 (in vitro) and 0–5 (hMSC). (**B**) In vitro ChIP-seq peaks total vs a 1 RPM threshold, plotted as a function of C/EBPβ titration (0.1, 0.25, 1 and 10 μl). Total C/EBPβ peaks found in hMSCs (red line) is included for point of comparison. (**C**) Box plot of peak strength in the 10 μl C/EBPβ cistrome comparing gained vs co-bound sites. The co-bound fraction met a 0.5 RPM threshold in all C/EBPβ homodimer cistromes. (**D**) Motif enrichment for the strongest- and weakest-1000 sites of the C/EBPβ cistromes. (**E**) Browser shots of C/EBPβ peaks upon titration of ATF (0.1, 1 and 10 μM) with C/EBPβ (60 nM). Tracks are RPM normalized, 0–3 RPM scale on the left panel, 0–20 RPM on the right. (**F**) K-means clustered density heat maps of C/EBPβ occupancy as a function of ATF4 titration. Titration of ATF4 (0.1, 1 and 10 μM, C/EBPβ IP; 1 and 10 μM, ATF4 IP) with C/EBPβ (60 nM) is indicated. ATF4 homodimer binding was examined

*Figure 6. continued on next page*

*Figure 6. Continued*

with 250 μM protein. Peaks thresholded to meet 1 RPM in at least two C/EBPβ cistromes. Three discrete clusters were identified and the de novo motif enrichment was based on all sites in each cluster.

The following figure supplement is available for figure 6:

**Figure supplement 1**. Characterization of binding associated with either in vitro-selective C/EBPβ sites or ATF4 homodimers.

---

Nevertheless, several candidate TFs have been identified in murine cell culture models. ZFP423 is expressed almost exclusively in adipogenic fibroblasts, and is necessary for adipocyte differentiation in vitro and for the development of adipose tissue in mice (*Gupta et al., 2010*). TCF7L1 promotes commitment of 3T3-L1 fibroblasts to a pre-adipocyte state in response to cell confluency (*Cristancho et al., 2011*). KAISO, in contrast, represses pre-adipocyte differentiation by recruiting the co-repressor SMRT to promoters (*Raghav et al., 2012*). Each of these studies marks an important advance in understanding pre-adipocyte biology, yet it remains unclear as to how, or if, these pre-adipocyte TFs functionally interact with each other or with the TFs responding to adipogenic signals. Here, the identification of the C/EBPβ-ATF4 heterodimer points to a mechanism that connects adipogenic competency and early differentiation through the sharing of C/EBPβ.

Mice lacking ATF4 are lean and resist diet-induced obesity, although they have adipose tissue (*Masuoka and Townes, 2002*; *Yang et al., 2004*; *Seo et al., 2009*). Our demonstration of a role for ATF4 in adult stem cell adipocyte differentiation suggest that this regulatory mechanism could be especially relevant in adult adipose tissue, consistent with the recent demonstration that adipogenesis occurs via distinct cell lineages during development vs adulthood (*Jiang et al., 2014*). Depletion of ATF4 in 3T3-L1 cells blocks adipocyte differentiation (*Yu et al., 2014*). ATF4 is also required for the development of blood and bone cell lineages (*Masuoka and Townes, 2002*; *Yang et al., 2004*; *Seo et al., 2009*). It is of interest to note that ATF4 plays a central role in handling ER stress induced by amino acid imbalance (*Harding et al., 2003*; *Kilberg et al., 2009*), and accumulating evidence indicates that ER stress pathways couple obesity to other metabolic dysfunctions such as diabetes (*Ozcan et al., 2004*; *Scheuner and Kaufman, 2008*), suggesting that ATF4 may regulate lipid and carbohydrate metabolism. Our observation that ATF4 regulates similar biological processes in hMSCs primed for differentiation may suggest an anticipatory response to meet an increased demand for protein synthesis during adipogenesis.

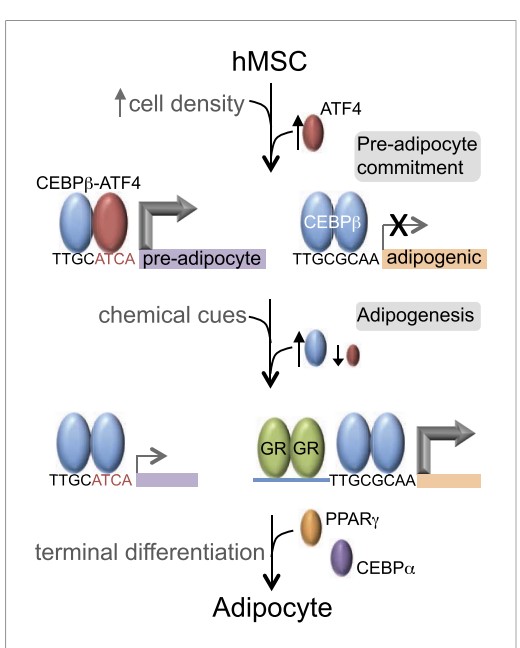

**Figure 7**. C/EBPβ controls distinct gene expression programs in hMSCs. Cell density regulates ATF4 expression to control C/EBPβ-ATF4 heterodimer formation. Prior to DMI treatment, heterodimer binding to hybrid motifs promotes a gene expression program priming hMSCs for adipogenesis. In this basal state, C/EBPβ is a poor transcriptional activator. In response to DMI, ATF4 level decreases, C/EBPβ level and GR activity increase. C/EBPβ initiates a program of transcription at new genes to orchestrate adipocyte differentiation with cooperating factors such as GR.

The binding profiles of ATF4 and C/EBPβ in hMSCs can be explained by the inherent sequence specificities of the homo- and heterodimeric complexes. First, genomic binding of ATF4 requires heterodimerization with C/EBPβ. While elegant biochemical studies have demonstrated the propensity of ATF4 and C/EBPβ to heterodimerize (*Vinson et al., 1993*), early studies also reported DNA-binding activity for ATF4 homodimers at palindromic cyclic AMP response

elements (CREs) (*Vinson et al., 1993*; *Podust et al., 2001*). In this light, our finding of a paucity of ATF4 homodimeric sites is unexpected, and indicates that while such interactions may be detectable when the relative concentration of the motif is high, the affinity of ATF4 homodimers for the CRE sequence is insufficient to produce binding in cells, or in vitro when a large amount of competitor DNA is present. In support of these coclusions, PBM-based assays failed to identify a robust motif for ATF4 homodimers (*Weirauch et al., 2014*), whereas HT-SELEX assays showed robust sequence preference of ATF4 for hybrid motifs (*Jolma et al., 2013*).

The IVC demonstrates that C/EBPβ and ATF4 target distinct DNA half sites, TTRS and TGAT, respectively. The relative plasticity of C/EBPβ homodimer in accommodating multiple DNA sequence variants is borne out in vivo, as the top motif associated with the cellular C/EBPβ cistrome shows considerable degeneracy at the −1 and −2 positions. In contrast, the C/EBPβ-ATF4 heterodimer is restricted to hybrid motifs of two inverted half-sites, TTRSATCA. Thus the biochemical properties of the heterodimer are sufficient to explain the nearly 1:1 correspondence between ATF4 binding and the presence of hybrid motifs in vivo. Interestingly, ATF4 heterodimerizes with the C/EBP related protein, CHOP, in the context of ER stress to bind hybrid motifs (*Han et al., 2013*), suggesting that while heterodimerization is required for DNA binding, the specific C/EBP partner can vary in different biological contexts.

C/EBPβ homodimers bind to hybrid sequences with low affinity relative to the C/EBPβ-ATF4 heterodimer, and this interaction requires elevated protein concentrations. DMI-treated hMSCs show persistent occupancy of the hybrid sites by C/EBPβ despite low ATF4 levels. An interesting possibility is that persistent binding is attributable to the elevated C/EBPβ concentrations in differentiating cells, however the effects of DMI-induced post-translational modification of C/EBPβ on homodimer binding cannot be excluded. Notably, robust RNAPII recruitment at hybrid sites was observed in cells only in the presence of ATF4. Thus, hetero vs homodimeric binding may have important consequences for licensing transcription activity, especially under conditions where the basal activity of C/EBPβ is limiting, possibly due to associated repressor complexes (*Wiper-Bergeron et al., 2003*; *Raghav et al., 2012*).

Our findings may have broad impact on the understanding of bZip TFs that readily form different dimer pairs (*Vinson et al., 2002*; *Newman and Keating, 2003*; *Reinke et al., 2013*), as well as TF families using other structural domains for the formation of homo- and heterodimeric protein complexes. By analogy with the C/EBPβ-ATF4 heterodimer, other C/EBPβ-bZip heterodimers may selectively recognize non-degenerate DNA sequences, which could illuminate fundamental rules governing their operation on a genome scale, and help to explain discrepancies between the palindromic C/EBP motif identified in vitro (*Jolma et al., 2013*; *Weirauch et al., 2014*) and the ChIP-seq motif with substantial variation at the central positions. Consistent with this paradigm, heterodimers displaying greatest affinity for TGACGCAA, which differs from the C/EBPβ-ATF4 motif by single T/C substitution at the fourth position, were observed through forced expression of C/EBPα with either c-JUN or c-FOS (*Hong et al., 2011*) and inferred from DNase I footprinting in 3T3-L1 cells (*Siersbæk et al., 2014a*). We propose that the degenerate C/EBP motif observed in cells results from the binding of different homo- and heterodimeric protein complexes with distinct sequence specificities. Because C/EBP proteins are widely expressed with additional bZip family members, determining how they target the genome is critical to understanding their control of basic processes including cell growth and development.

## Materials and methods

### Experimental procedures
#### Cell culture and reagents
hMSCs were obtained from Lonza and maintained in low glucose Dulbecco's modified Eagle's medium containing supplemented with 10% fetal bovine serum and 2 mM glutamine. hMSCs were used at passages 4 through 7. For cell density manipulations, hMSCs were re-plated at low (2000 cells/cm$^2$) or high (30,000 cells/cm$^2$) densities. For differentiation studies, DMI medium was added to cells that had been cultured overnight at the desired cell density. DMI formulation was as follows: DMEM with 10% FBS, 2 mM glutamine, 1 μM dexamethasone, 0.5 mM IBMX, and 0.2 mM indomethacin, and 10 μg/ml insulin. 3T3L1s and C3H10T1/2 cells were obtained from ATCC.

## Luciferase reporter assays

Candidate enhancers were amplified from hMSC genomic DNA by touchdown PCR, using oligos that added 5′ XhoI and 3′ BamHI restriction sites. Amplified DNA was gel isolated, digested, and ligated into pGL4.24. Genomic coordinates for candidate enhancers are reported in the Supplemental Information. C310T1/2 cells were co-transfected with 1.8 mg of luciferase reporter construct and 0.2 mg of CMV-Renilla plasmid using the TransIT LT1 transfection reagent. Following an overnight transfection, cells were re-plated at high cell densities (50,000 cells/cm$^2$) in duplicate wells. Cells were then treated with DMI medium for an addition 24 hr followed by treatment with passive lysis buffer. Relative luminescence (Firefly to Renilla ratio) was determined on a Glomax luminometer. hg19 genomic coordinates for each reporter are as follows: ASNS, chr7:97501555-97502014; MTHFD2, chr2:74426247-74426355; SLC7A5, chr16:87887210-87887550; Intergenic hybrid, chr1:204476231-204476350; control C/EBPβ (LAMβ1), chr7:107598937-107599756; ASTN2, chr9:119212591-119213520; HSD11b1, chr1:209893618-209894063; PPARγ2, chr3:12391004-12392991. Oligos for subcloning candidate enhancers are listed in the *Supplementary file 2*.

## ATF4 knockdown, mRNA analysis and western blotting

ON-TARGET siRNA duplexes were purchased from Dharmacon/Thermo Scientific. hMSCs were transfected with RNAi duplexes or non-targeting siGLO at 200 μM final concentration using Lipofectamine RNAiMAX. 3T3L1s were electroporated as described previously (*Cristancho et al., 2011*). Cells were cultured for 48–72 hr following transfection prior to experimental manipulation. For mRNA analysis, RNA was isolated using the Qiagen RNeasy micro kit. cDNA synthesis on 0.5–1.0 μg of total RNA was performed using MultiScribe reverse transcriptase (Life Technologies, Carlsbad, CA). RAD17 was used as an endogenous control for normalization. qPCR was performed using Power SYBR Green PCR Master mix (Life Technologies). Primers are listed in *Supplementary file 2*. For western analyses, hMSCs or 3T3L1 extracts were prepared by direct lysis of cells in 1× Laemmli buffer for western blotting. Extracts were resolved on 10% Tris-glycine gels and transferred to PVDF. C/EBPβ and ATF4 (mAb D4B8, Cell Signaling) antibodies were used at a 1:1000 dilution, anti-RAN (clone 20/Ran, BD Biosciences, San Jose, CA) at a 1:10,000 dilution, HRP-conjugated anti-rabbit and anti-mouse at 1:10,000. ECL was performed with SuperSignal West Femto.

## EMSA and IVC

Binding reactions were performed in 20 μl (EMSA) or 100 μl (IVC) with 10 mM HEPES pH 7.8, 50 mM NaCl, 10 mM MgCl$_2$, 5 mM DTT and Complete protease inhibitor. EMSA reactions also contained 0.25 mg/ml BSA, 5 ng/μl poly dIdC, 5% glycerol and ~5 fmole $^{32}$P-radiolabeled ds oligonucleotide. IVC reactions contained 1 mg/ml BSA and 500 ng of isolated hMSC genomic DNA fragmented to an average size of ~150 bp. hMSC DNA was isolated using DNAzol (Invitrogen, Carlsbad, CA), further stripped of protein by Proteinase K treatment and phenol/chloroform extraction, and sheared by sonication with a Covaris M220 focused ultrasonicator. ATF4 and C/EBPβ were expressed in BL21(DE3) *E. coli* from pET30a vectors carrying an N-terminal His Tag and full-length cDNAs. Proteins were purified by Co$^{2+}$ affinity chromatography and quantified by comparison with a BSA standard after SDS-PAGE. EMSA binding reactions were resolved by native 6% PAGE, with 10 mM MgCl$_2$ in the gel to increase the DNA-binding specificity of bZip proteins (*Moll et al., 2002*), and were imaged and quantified using a Typhoon 9400 Imager. IVC binding reactions were brought to 1 ml with binding buffer lacking DTT for immunoprecipitation. Library preparation and sequencing were performed similarly to ChIP-seq.

## ChIP and ChIP-Seq

The following antibodies were used for ChIP: C/EBPβ (sc-150, Santa Cruz Biotechnology, Dallas, TX), GR (PA1-511A, Pierce, sc-1004, Santa Cruz), ATF4 (sc-200, Santa Cruz), RNAPII (sc-899, sc-9001, Santa Cruz), H3K27ac (ab4729, Abcam, Cambridge, MA), ATF2 (sc-6233, Santa Cruz), ATF-3 (sc-188, Santa Cruz), C/EBPg (sc-25769, Santa Cruz), GADD153 (sc-575, Santa Cruz), FOS (sc-7202, Santa Cruz), FOSL1 (sc-183, Santa Cruz), JUND (sc-74, Santa Cruz), and MAFF/G/K (sc-22831, Santa Cruz). The first IP for sequential ChIP was performed with whole cell extract derived from approximately 5 million formaldehyde-crosslinked hMSCs. After overnight incubation at 4°C with antibody and protein G sepharose in 1 ml ChIP buffer (50 mM HEPES pH 7.8, 140 mM NaCl, 1% Triton X-100, 0.1% Na-Deoxycholate and Complete protease inhibitor), IPs were washed and then eluted by incubation at 65°C for 15 min in 50 μl ChIP elution buffer (50 mM Tris pH 7.5, 10 mM EDTA, 1% SDS) with 10 mM DTT. Beads were pelleted and the supernatant transferred to a new tube. A subsequent wash with

50 µl ChIP elution buffer was performed with the beads, and the supernatants combined. Eluted protein-DNA complexes were added to 1 ml ChIP buffer with 5 mg/ml BSA and 2 µg nonspecific DNA. Samples were split into two equal parts and volumes raised to 1 ml with ChIP buffer. One aliquot received antibody against a particular TF, the other rabbit IgG, and samples were incubated at 4°C for 1–2 hr. Immune complexes were captured by protein G sepharose, washed, eluted, and incubated overnight at 65°C to reverse formaldehyde crosslinks. DNA was subsequently isolated for PCR. Primers are listed in *Supplementary file 2*.

ChIP-Seq libraries were prepared from formaldehyde-crosslinked hMSCs, using approximately 2–8 million cells per condition, depending on the abundance of the factor of interest. Libraries were sequenced as previously described (*Steger et al., 2010*). Analyses of ChIP-Seq tracks were performed using uniquely aligned reads that were converted to unique stack height profiles (USHPs) using the GenomeCoverageBed algorithm (BEDTools, Galaxy genomics suite). A normalization factor was applied to the USHP data to account for differences in unique reads between ChIP-seq tracks prepared for more than one experimental condition (i.e., RNAPII and C/EBPβ). Peak calling on TF ChIP-Seq tracks and all de novo motif analyses were performed using HOMER (*Heinz et al., 2010*). Peak calling for RNAPII and sequence tag analyses for features of interest were performed on the Galaxy and Cistrome servers. CEAS analysis (*Shin et al., 2009*) and generation of Venn Diagrams were performed on Cistrome. Mapping RNAPII to nearby genes was done using GREAT (*McLean et al., 2010*). GO analysis was conducted with DAVID Resources (*Dennis et al., 2003*). Average profile plot and box plots were generated using R. Outliers on box plots (exceeding 1.5× interquartile distance) were excluded.

## Computational analyses of genomics data

### RNAPII track processing

RNAPII data was windowed into 20 bp bins, and a USHP measured for each bin. To eliminate artifacts introduced by regions of low ChIP signals caused by poor mappability, regions of no RNAPII sequence coverage were masked based on their mappability scores from the ENCODE Duke Uniqueness 35 bp track. Masked regions (low mappability, low ChIP signal) that were juxtaposed to bins with enriched RNAPII signal were assigned an interpolated USHP score, that is, the average USHP score of adjacent upstream and downstream bins unaffected by mappability. RNAPII data were smoothened by recalculating USHP scores as the moving average of a 60 bp window centered on each 20 bp bin. Genomic regions with high sequencing depth in the input (no ChIP) tracks were subtracted from the RNAPII data to remove false positive peaks. Finally, each bin was assigned a 'USHP slope' score computed from the difference of the USHP values for the bin lying 60 bp downstream vs the bin 60 bp upstream divided by the intervening bp length (100 bp internally).

### RNAPII feature detection

RNAPII features were defined as the intervals spanning regions of increasing USHP slopes paired to regions of decreasing USHP slopes. Briefly, RNAPII bins were thresholded for USHP slope scores and clustered to detect regions of positive or negative USHP slopes. Each positive sloping region was joined with the nearest downstream negative slope region, with a maximum cutoff distance of 400 bp between the 3′ and 5′ ends of the positively and negatively sloping regions. Adjacent RNAPII features were then evaluated for peak merging based on the peak to trough ratios (rations < 2.5 were merged), such that complex RNAPII features comprised of poorly resolved multiplets were treated as a single feature. To evaluate enrichment of sequence reads within the putative RNAPII features, features within 2 kb were clustered, and these domains expanded to ±1 kb to sample the local RNAPII background. Peak-associated bins within these domains were masked and the average USHP signal per bin in the local neighborhood computed. RNAPII features with USHP maxima that were 3.5-fold greater than the mean of non-peak associated local USHP bins were used in subsequent analyses. In addition, RNAPII features intervals were subject to a signal threshold requiring a tag count of ≥20 per feature or a tag density of at least 14.5 sequence tags per bp. All steps of the RNAPII feature detection algorithm were scripted through the Galaxy server, and will be available as published workflows.

### Heat map clustering

For every RNAPII peak, tag counts were determined across all four experimental tracks (HD vs LD, with or without DMI treatment). Each peak was assigned attributes to indicate whether RNAPII exhibited a twofold or greater change in occupancy for each pairwise comparison of experimental conditions, with three

possible attributes per comparison (up, no change, or down in response to density or DMI manipulations). Following attributed-based group assignment, the complexity of the clustering results was simplified by grouping together functionally related clusters. Similar classes of RNAPII were detected with non-supervised clustering methods (k-means clustering using Cluster 3.0). For heat map representations of the RNAPII data, the data was transformed to represent the $\log_2$ fold change of the condition-specific tag count relative to the mean tag count observed for all four RNAPII tracks (e.g., $\log_2$ [tags in condition #1/mean tag count at the RNAPII peak for all conditions]). Heat maps were generated in TreeView (*Saldanha, 2004*).

## De novo motif analysis on RNAPII peaks

For each peak interval, motif analysis was centered on the bins scoring in the top 25th percentile of USHP values observed within the peak interval. A minimum window size of 400 bp around these top 25th percentile bin was used for motif mining. Windows around the top 25th percentile of bins were limited to a maximum length of 600 bp (peaks that failed this threshold were excluded). Overlapping windows (due to the detection of a given RNAPII peak call in more than one ChIP-Seq track) were merged into single intervals. Motif mining was performed with HOMER using the hg19r genome build and the 'size given' peak option. Sequence motifs reported in *Figure 1C* were limited to those found by HOMER in at least 5% of sites and with p-values $\leq$ 1e-14.

## Supplemental information

Supplemental figures and files are available. Genome-wide data sets are available at the NCBI Gene Expression Omnibus (GEO; http://www.ncbi.nlm.nih.gov/geo/) under accession number GSE68864.

## Acknowledgements

We thank Alex Ruan for technical assistance, Dr Jared Bushman for kindly providing the luciferase protocol for D-luciferin and h-Coelenterazine substrates, and members of the Lazar and Chen laboratories for helpful discussions. We thank Drs Costas Koumenis and Celeste Simon for providing ATF4-specific siRNAs and antibodies. We also thank the Functional Genomics Core of the Penn Diabetes Center (DK19525) for deep sequencing. This work was supported by NIH grants 1F32 HL084966 (to DMC), R21 DK098769 (to K-JW), R01 DK49780 (to MAL), R01 GM074048 (to CSC) and R01 DK098542 (to DJS).

## Additional information

### Funding

| Funder | Grant reference | Author |
| --- | --- | --- |
| National Institutes of Health (NIH) | 1F32 HL084966 | Daniel M Cohen |
| National Institutes of Health (NIH) | R21 DK098769 | Kyoung-Jae Won |
| National Institutes of Health (NIH) | R01 DK49780 | Mitchell A Lazar |
| National Institutes of Health (NIH) | R01 GM074048 | Christopher S Chen |
| National Institutes of Health (NIH) | R01 DK098542 | David J Steger |

The funder had no role in study design, data collection and interpretation, or the decision to submit the work for publication.

### Author contributions

DMC, DJS, Conception and design, Acquisition of data, Analysis and interpretation of data, Drafting or revising the article; K-JW, NN, Analysis and interpretation of data, Drafting or revising the article; MAL, CSC, Conception and design, Analysis and interpretation of data, Drafting or revising the article

## Additional files

### Supplementary files

• Supplementary file 1. Gene Lists Associated with Gene Ontology Analysis, Related to *Figure 1*. RNAPII regions were mapped to nearby genes using GREAT (*McLean et al., 2010*). RNAPII occupancy

was scored for each gene as a function of cell density and DMI treatment and genes showing correlated changes between an RNAPII peak and the entire gene locus were submitted to DAVID (*Dennis et al., 2003*) for Gene Ontology (GO) analysis (Panther and KEGG Pathways). Spreadsheets list RNAPII differential regions mapped to genes ('by Enhancer'), with distance to the transcription start site (TSS), and accompanying GO Terms and RNAPII cluster designation (see *Figure 1A*), as well as contain summary lists of all differential regions associated with each gene ('by Gene'), and all genes associated with each GO Term ('by GO Term').

• Supplementary file 2. Oligonucleotides Used in the Study. Spreadsheets list PCR primers used to subclone candidate enhancers for luciferase assays, to examine first-strand cDNA for mRNA expression, and to interrogate ChIP DNA for TF occupancy.

### Major datasets
The following dataset was generated:

| Author(s) | Year | Dataset title | Dataset ID and/or URL | Database, license, and accessibility information |
|---|---|---|---|---|
| Cohen D, Steger D | 2015 | ATF4 Licenses C/EBPb Activity in Human Mesenchymal Stem Cells Primed for Adipogenesis | http://www.ncbi.nlm.nih.gov/geo/query/acc.cgi?acc=GSE68864 | Publicly available at the NCBI Gene Expression Omnibus (Accession no: GSE68864). |

The following previously published datasets were used:

| Author(s) | Year | Dataset title | Dataset ID and/or URL | Database, license, and accessibility information |
|---|---|---|---|---|
| ENCODE Consortium | 2013 | wgEncodeRegTfbsClusteredV3.bed.gz | http://hgdownload.cse.ucsc.edu/goldenPath/hg19/encodeDCC/wgEncodeRegTfbsClustered/ | Publicly available at hgdownload.cse.ucsc.edu. |
| ENCODE Consortium | 2013 | wgEncodeRegDnaseClusteredV2.bed.gz | http://hgdownload.cse.ucsc.edu/goldenPath/hg19/encodeDCC/wgEncodeRegDnaseClustered/ | Publicly available at hgdownload.cse.ucsc.edu. |
| ENCODE Consortium | 2011 | Open Chromatin by DNaseI HS from ENCODE/OpenChrom | http://www.ncbi.nlm.nih.gov/geo/query/acc.cgi?acc=GSE32970 | Publicly available at NCBI Gene Expression Omnibus (GSE32970). |
| ENCODE Consortium | 2011 | Transcription Factor Binding Sites by ChIP-seq from ENCODE/HAIB | http://www.ncbi.nlm.nih.gov/geo/query/acc.cgi?acc=GSE32465 | Publicly available at NCBI Gene Expression Omnibus (GSE32465). |

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
