## [Decision Letter]

Thank you for sending your work entitled “ATF4 licenses C/EBPβ activity in human mesenchymal stem cells primed for adipogenesis” for consideration at *eLife*. Your article has been favorably evaluated by Fiona Watt (Senior editor) and 3 reviewers, one of whom, Stephen Farmer, has agreed to share his identity.

The editor and the reviewers discussed their comments before we reached this decision, and the editor has assembled the following comments to help you prepare a revised submission.

All the reviewers considered your studies to be well executed. However, a major concern is the biological relevance of the reported discovery, in particular the role of cell density or confluence as a “trigger” during adipogenesis in vivo. If the authors would like to make the argument that this mechanism relates specifically to adipogenesis their data are incomplete. ATF4 knockout mice have adipose tissue although they are leaner and resist diet induced obesity, but there is little evidence they have a defect in adipogenesis. It would appear from a number of studies that ATF4 plays a more important role in regulating energy metabolism than in directly affecting adipogenesis. In fact, the authors never directly assess adipogenesis but rather lipid accumulation, which can be affected by altering cellular metabolism. The authors should attempt to distinguish between the two: it would be relatively straightforward to examine the cells metabolically, for instance flux analysis, metabolomics, simple glycolysis assays. Without some attempt to distinguish between a differentiation phenotype and a cellular metabolic phenotype many of the conclusions about adipogenesis are not supported by the data. Furthermore, in addition to measuring oil droplet accumulation a qPCR survey of dynamic adipocyte differentiation marker expression should be conducted to elucidate the block in greater detail.

A previous study by Yu et al. (cited in the manuscript) has suggested that ATF4 controls expression of C/EBPβ and PPARγ through promoter binding in a 3T3 model. The current manuscript suggests a model where ATF4 binding at enhancer elements is critical. Can the authors distinguish these two models experimentally, e.g. through rescue of adipogenesis in ATF knockdown cells through C/EBPβ overexpression?

The authors describe 1451 ATF4 ChIP-seq peaks (Figure 3) that are found in HD MSC cells. Presumably, the peaks fall mainly within cluster 10-11 from Figure 1 since there is an enrichment of the hybrid motif, but it is hard to follow if the ATF4 binding events also exhibit a certain GO enrichment which could clarify the biological context ATF4 binding (or absence thereof) might have. The manuscript might greatly benefit from a clearer analysis and display of ATF4-binding events/hybrid motif usage.

---

## [Author Response]

*All the reviewers considered your studies to be well executed. However, a major concern is the biological relevance of the reported discovery, in particular the role of cell density or confluence as a “trigger” during adipogenesis in vivo. If the authors would like to make the argument that this mechanism relates specifically to adipogenesis their data are incomplete. ATF4 knockout mice have adipose tissue although they are leaner and resist diet induced obesity, but there is little evidence they have a defect in adipogenesis. It would appear from a number of studies that ATF4 plays a more important role in regulating energy metabolism than in directly affecting adipogenesis*.

We appreciate the reviewers’ concerns. We have made the following modification to the second paragraph of the discussion to provide a reference point for readers: This study identifies ATF4 as a cell density sensor that links the physical and chemical requirements for adipogenesis. High cell density has long been known to be an important condition for adipocyte differentiation in vitro (7; 12; 33; 40). It is likely that a combination of cues is required in tissue microenvironments supporting adipogenesis in vivo.

Regarding adipogenesis in ATF4 knockout mice, we agree and make the following modification to the Discussion’s third paragraph: Mice lacking ATF4 are lean and resist diet-induced obesity, although they have adipose tissue (32; 51; 68). Our demonstration of a role for ATF4 in adult stem cell adipocyte differentiation suggest that this regulatory mechanism could be especially relevant in adult adipose tissue, consistent with the recent demonstration that adipogenesis occurs via distinct cell lineages during development versus adulthood (22).

*In fact, the authors never directly assess adipogenesis but rather lipid accumulation, which can be affected by altering cellular metabolism. The authors should attempt to distinguish between the two: it would be relatively straightforward to examine the cells metabolically, for instance flux analysis, metabolomics, simple glycolysis assays. Without some attempt to distinguish between a differentiation phenotype and a cellular metabolic phenotype many of the conclusions about adipogenesis are not supported by the data. Furthermore, in addition to measuring oil droplet accumulation a qPCR survey of dynamic adipocyte differentiation marker expression should be conducted to elucidate the block in greater detail*.

We thank the reviewers for raising this point; clarifying how ATF4 affects lipid accumulation in response to adipogenic signals improves the study. To directly address adipogenesis, we measured the expression of key differentiation markers in hMSCs with normal and reduced levels of ATF4. Transcriptional activation of *PPARγ* and *C/EBPα*, both encoding adipose-determining transcription factors, is significantly decreased in hMSCs depleted for ATF4 in response to adipogenic signals. Several downstream gene targets important for adipocyte differentiation also have reduced expression, suggesting that ATF4 affects lipid accumulation by promoting adipogenesis. These data have been added to Figure 4, and referred to in the subsection headed “Loss of ATF4 Attenuates Adipogenesis and Cell-Density-Dependent Transcription” as follows: In parallel, early markers of adipogenesis were decreased 3-4-fold in ATF4 knockdown cells (Figure 4), including the mRNAs encoding the lineage-determining TFs PPARγ2 and C/EBPα, as well as several of their downstream transcriptional targets. This deficit in adipogenic gene expression became more pronounced over time (Figure 4—figure supplement 1), and together, these data suggest that ATF4 expression is important for adipogenesis.

*A previous study by Yu et al. (cited in the manuscript) has suggested that ATF4 controls expression of C/EBPβ and PPARγ through promoter binding in a 3T3 model. The current manuscript suggests a model where ATF4 binding at enhancer elements is critical. Can the authors distinguish these two models experimentally, e.g. through rescue of adipogenesis in ATF knockdown cells through C/EBPβ overexpression*?

Our data do not indicate that ATF4 directly regulates *C/EBPβ* or *PPARγ* expression. Both mRNA and protein for C/EBPβ appear unchanged in hMSCs depleted of ATF4 (Figure 8). It should be noted that Yu et al. report ATF4 binding by ChIP to cyclic AMP response elements (CREs) in the *Cebpβ* (-152 to 0) and *Pparγ* (-360 to -58) promoters as evidence for direct regulation of these genes, yet ATF4 displays little or no interaction with the CRE motif in unbiased genome-wide studies from mouse embryonic fibroblasts (Nature Cell Biology 2013. doi:10.1038/ncb2738) and hMSCs (this work).

Author response image 1.(A) C/EBPβ gene expression in HD hMSCs treated with control (PPIB) or ATF4 siRNAs. Error bars depict SEM. As shown in Figure 4, ATF4 mRNA is dramatically reduced by the treatment. (B) ATF4 knockdown in HD hMSCs assessed by western blotting for two biological replicates. RAN, loading control.**DOI:**
http://dx.doi.org/10.7554/eLife.06821.020

Furthermore, we do not observe binding at either region in hMSCs (Figure 9). A single ATF4 peak resides at the *PPARγ* locus, 40 kb downstream of the *PPARγ*2 TSS, yet it lacks enrichment for either RNAPII or H3K27ac in high-density-seeded cells, suggesting little or no enhancer activity under these conditions

Author response image 2.CEBPβ (green) and ATF4 (yellow) ChIP-seq tracks in HD hMSCs. Tracks are RPM normalized and presented with the same y-axis scale (0 - 5).**DOI:**
http://dx.doi.org/10.7554/eLife.06821.021

We appreciate the suggestion to distinguish between conflicting models. However, rather than restoring function, forced expression of C/EBPβ under ATF4 knockdown conditions will yield abnormally high C/EBPβ levels, limiting the interpretation of any potential effects on adipogenesis. Forced C/EBPβ expression has been shown by the Spiegelman and Farmer labs to trigger adipogenesis in NIH-3T3 cells, likely by overwhelming and/or overriding the system and obviating the need for pre-adipocyte-specific transcription. Thus, we do not believe that this experiment can resolve how ATF4 contributes to adipogenic commitment.

*The authors describe 1451 ATF4 ChIP-seq peaks (*Figure 3*) that are found in HD MSC cells. Presumably, the peaks fall mainly within cluster 10-11 from*
Figure 1
*since there is an enrichment of the hybrid motif, but it is hard to follow if the ATF4 binding events also exhibit a certain GO enrichment which could clarify the biological context ATF4 binding (or absence thereof) might have. The manuscript might greatly benefit from a clearer analysis and display of ATF4-binding events/hybrid motif usage*.

We thank the reviewers for these suggestions, as they have led to an improved study. GO results derived from mapping ATF4-binding sites to nearby genes are now included in Figure 3—figure supplement 1. They are similar to the GO results obtained from the HD-primed RNAPII sites. To provide a clearer display of the relationship between ATF4 and RNAPII genomic occupancy, we determined the fraction of ATF4 or C/EBPβ sites that co-localize with RNAPII, and present the results as pie charts in Figure 3—figure supplement 1. Description of these data has been added the following to the second paragraph of the subsection headed “ATF4 Is a Density-Dependent Factor that Targets HD-primed Enhancers as a Heterodimer with C/EBPβ”: “Genomic occupancy of ATF4 is associated predominantly with genes […] in response to high-cell-density seeding of hMSCs.”

We have also expanded the third paragraph of the Discussion based on these results as follows: “It is of interest to note that ATF4 plays a central role […] demand for protein synthesis during adipogenesis.”